# Mathematical Modeling and Microparticle Size Control for Enhancing Heat Transfer Efficiency in High-Viscosity Food Suspensions

**DOI:** 10.3390/foods14152625

**Published:** 2025-07-26

**Authors:** Hyeonbo Lee, Mi-Jung Choi, Jiseon Lee

**Affiliations:** 1Department of Food Science and Biotechnology of Animal Resources, Konkuk University, Seoul 05029, Republic of Korea; ben1900@naver.com; 2School of Animal, Food Science and Marketing, Konkuk University, Seoul 05029, Republic of Korea; choimj@konkuk.ac.kr

**Keywords:** microparticle size, heat transfer efficiency, high-viscosity suspension, Rayleigh–Bénard convection, thermal conductivity, rheology, Herschel–Bulkley

## Abstract

This study investigated how microparticle size affects natural convective heat transfer in high-viscosity suspensions. Suspensions were formulated using 0.5% xanthan gum and 3% stearic acid, with particle sizes ranging from 120 to 750 nm. Key thermal properties, including thermal conductivity (0.598–0.679 W/m·K), specific heat, and the volumetric thermal expansion coefficient (0.990–1.000/°C), were measured. Rheological analysis based on the Herschel–Bulkley model revealed that reducing the particle size increased the consistency index from 0.56 to 0.75 Pa·s, while reducing the flow index from 0.63 to 0.50. This indicates enhanced shear-thinning behavior. A Rayleigh–Bénard convection system revealed that suspensions containing smaller particles exhibited higher Rayleigh and Nusselt numbers under large temperature gradients. Nusselt numbers reached values of up to 100 at a temperature difference of 9 °C. Conversely, suspensions containing larger particles exhibited relatively higher Rayleigh and Nusselt numbers under smaller temperature differences. These results demonstrate that optimizing microparticle size can enhance the efficiency of heat transfer in high-viscosity suspensions depending on the applied thermal gradient. This has practical implications for improving heat transfer in food and other viscous systems where convection is limited.

## 1. Introduction

One common approach to enhancing heat transfer involves the use of nanoparticles (particle size: 1–100 nm) or microparticles (particle size: 1–1000 µm) [1,2]. Micro- and nanoparticles (hereafter referred to as micro-nanoparticles) can either be composed of materials with high thermal conductivity or can increase the number of heat transfer pathways by expanding the surface area because of their small size, thereby facilitating more efficient heat movement within the fluid [2]. Microparticles lack some of the mechanisms that drive thermal improvements in nanofluids, such as Brownian motion or significantly increased surface area [3]. Additionally, microparticles exhibit lower dispersion stability than nanoparticles, resulting in issues such as sedimentation over time [3]. While nanoparticles offer superior thermal performance, their typical composition, metal or carbon-based materials, often limits their suitability for human-compatible applications, such as in the food and pharmaceutical industries [4].

Efforts to develop nanoparticles using natural polymers have faced challenges, including concerns over toxicity, high production costs, and instability during mass production [5]. Although nanoparticles have been widely studied for enhancing heat transfer in food engineering, most of these studies have focused on external heating media rather than direct dispersion into food matrices. Research on applying micro- or nano-sized particles to improve convective heat transfer in actual food systems remains limited [6,7]. In contrast, microparticles offer greater physical stability, reducing the risk of deformation and reactivity. Additionally, microparticles can be produced using organic materials, such as proteins, fatty acids, and polymers, making them easier and more cost-effective to manufacture, and thus more suitable for large-scale production [8]. Furthermore, in the case of high-viscosity continuous phases, it is possible to enhance the dispersion stability of microparticles, which typically exhibit lower dispersion stability compared to nanoparticles [3]. Based on these considerations, stearic acid and xanthan gum were selected as representative dispersed and continuous phases, respectively, for constructing a food-compatible microparticle suspension system. Stearic acid was chosen for its well-defined physicochemical properties, such as a high melting point (~68–69 °C) and latent heat, which make it suitable for modeling heat transfer. Its ability to form uniform crystalline microparticles supports particle size control, and its food-grade status ensures compatibility with food systems [9,10]. Xanthan gum, an anionic microbial polysaccharide, was selected for the continuous phase because of its excellent suspension-stabilizing capacity. It forms a weak network structure at rest through intermolecular interactions, which easily breaks down under shear. This shear-thinning behavior makes it suitable for modeling high-viscosity food matrices under dynamic thermal processing conditions [11].

Few studies have focused on microparticles in high-viscosity fluids to enhance the natural convection heat transfer efficiency or improve thermal properties. Research is even more limited when the thermal conductivity of the dispersed phase is lower than that of the continuous phase. This study aims to verify the potential for improving the thermal properties and natural convection heat transfer efficiency of suspensions by controlling the size of microparticles with lower thermal conductivity than the continuous phase. These advancements could enhance processing efficiency, product quality, and energy savings in industries utilizing high-viscosity suspensions with microparticles.

## 2. Materials and Methods

### 2.1. Materials

Tween^®^ 80 (polyoxyethylene (20) sorbitan monooleate), which is a non-toxic and biocompatible surfactant, and xanthan gum (reported as 100% purity in the safety data sheet) were procured from Daejung (Siheung, Republic of Korea) for suspension preparation. Insulation material, extruded polystyrene (XPS; Isopink, Daewon Woodboard, Seoul, Republic of Korea), was purchased from a local market and used for thermal conductivity measurement.

### 2.2. Analytical Sample Determination

#### 2.2.1. Preparation of Stearic Acid Suspension

The stearic acid suspensions were prepared using an emulsion-based particle synthesis method [1,12]. The stearic acid concentration was fixed at 3 wt.% to model the fat content typically found in high-viscosity food suspensions such as curry sauces, which often contain 3–7 wt% fat. The overall suspension system was designed to represent thickened, flowable food products such as sauces, where non-Newtonian flow behavior and dispersed lipid phases are common. This concentration was chosen to ensure experimental control and maintain practical applicability in terms of fat content. The four suspensions were prepared by stirring in 0.6, 1, 3, and 3% Tween^®^ 80 solutions at 30, 1780, 7120, and 28,620× *g*, respectively. The stearic acid was mixed with the prepared Tween^®^ 80 aqueous solution at 6 wt%. The mixed samples were double-boiled at 80 °C using a hot plate. During double boiling, the samples were homogenized for 5 min using a propeller stirrer (Chang Shin Scientific Co., Ltd., Seoul, Republic of Korea) or high-speed homogenizer (T25 digital Ultra-Turrax, IKA, Staufen, Germany). Ice was mixed with the homogenized samples at 50 wt% to crystallize the stearic acid oil into fat. The crystallization of the samples was completed by mixing an equal amount of ice and 2 wt% xanthan gum aqueous solution and stirring at 71× *g* for 5 min. Samples not mixed with 6 wt% Tween^®^ 80 were concentration-corrected by adding Tween^®^ 80 to 3 wt% of the final sample weight, along with the xanthan gum solution. A control sample was prepared in the same manner, without the addition of stearic acid, to confirm the effect of the presence or absence of particles.

#### 2.2.2. Particle Size and Size Distribution

The particle sizes of the suspension samples were measured using a laser diffraction spectrometer (Mastersizer 3000E, Malvern, London, UK), following the method described by Kwak et al. [12]. The aqueous phase and stearic acid refractive indices were set to 1.33 and 1.63, respectively. Stearic acid was added to the suspension until an obscuration of approximately 10% was obtained. The volume-weighted mean D_[4,3]_ was used as the average particle size of the suspension. The span value (i.e., the smaller the value, the closer *d*_10_ and *d*_90_ are and the narrower the particle distribution) is represented as the distribution. The span was calculated using the following Equation (1):(1)$$Span\;value=\frac{d_{90}-d_{10}}{d_{50}}$$
where *d*_10_, *d*_50_, and *d*_90_ are the equivalent volume diameters at 10%, 50%, and 90% of the cumulative diameter, respectively.

For samples with particle sizes below the measurement range of the laser diffraction spectrometer, measurements were carried out using a Zetasizer (Nano ZS90, Malvern, London, UK) that utilizes dynamic light scattering, according to the method described by Choi et al. [13]. The samples were diluted 200-fold in distilled water for the measurement. The suspension particle size distribution was expressed as PDI, a dimensionless number indicating the polydispersity within a sample. PDI was calculated using the following Equation (2):(2)$$PDI={\left(\sigma /D\right)}^{2}$$
where *σ* and *D* represent the standard deviation and mean of particle size, respectively. All samples were measured in triplicates at 25 °C.

### 2.3. Optical Characterization

The suspension samples were placed on slide glasses covered with cover glasses, and the particle morphology was observed using a polarized light microscope (Eclipse LV100NDA; Nikon Instruments Inc., Tokyo, Japan).

### 2.4. Rheological Properties

The viscosities of the stearic acid suspensions were measured using a rheometer (MCR302, Anton Paar, Graz, Austria). The sample was placed under a plate–plate probe (PP50; diameter: 50 mm, gap: 0.3 mm) before trimming the upper plate to fit within the probe. A thin layer of silicone oil was applied to the edges to prevent evaporation of the sample. Xanthan gum is a thickening agent with minimal trends in rheological changes across varying temperatures. Therefore, its rheological properties were measured only at 25 °C, with the highest average molecular weight [14]. The sample was maintained at 25 °C and stirred at an increasing shear rate of 0.005–1000 1/s. The shear stress measured alongside the viscosity was applied to estimate the shear rate dependence of the viscosity of the samples using the Herschel–Bulkley model. The measured shear stress and shear rate data were fitted to the Herschel-Bulkley model equation using OriginPro software version 2021 (OriginLab Corp., Northampton, MA, USA) as shown in Equation (3):(3)$$\tau =\tau_{0}+K{\dot{\gamma }}^{n}$$
where *τ*, *τ*_0_, *K*, $\dot{\gamma }$, and n represent shear stress (Pa), apparent yield stress (Pa), consistency index (Pa·s^n^), shear rate (1/s), and flow index, respectively.

To measure the viscoelastic modulus of stearic acid suspension samples, a rheometer (MCR302) was used. The sample was placed under the parallel-plate probe (PP25) and the gap was set to 1 mm before trimming the upper plate to fit within the probe. A thin layer of silicone oil was applied to the edges to prevent sample evaporation. The temperature sweep test was performed from 20 °C to 50 °C with a heating rate of 2 °C/min. The storage (G′) and loss modulus (G″) were measured within the identified linear viscoelastic region at 1.0 Hz frequency, and 0.1% shear strain.

### 2.5. Thermal Properties

#### 2.5.1. Specific Heat Capacity

Differential scanning calorimetry (DSC; DSC 200; Netzsch, Waldkraiburg, Germany) was used to perform specific heat capacity measurements. A sapphire standard established the specific heat capacity reference [15]. The temperature range was set between 10 °C and 40 °C, with a heating rate of 3 °C/min. The temperature range was determined by considering the melting point of the dispersed phase, stearic acid [16]. The obtained DSC thermograms were analyzed using Proteus version 9 (Netzsch, Waldkraiburg, Germany) to determine the specific heat capacity of the samples.

#### 2.5.2. Thermal Conductivity

The thermal conductivity was calculated using the following Equation (4):(4)$$k=\alpha \;\times \;C_{p}\;\times \;\rho$$
where *α*, *C_p_*, and *ρ* represent thermal diffusivity (m^2^·s^−1^), specific heat capacity (J·g^−1^·C^−1^), and density, respectively.

Thermal diffusivity of stearic acid suspension was measured using a light flash apparatus (LFA; LFA 467 HyperFlash, Netzsch, Waldkraiburg, Germany). Before measurement, the instrument was calibrated using an aluminum plate coated with graphite. The stearic acid suspensions were composed of a three-layer structure consisting of an aluminum plate, sample, and aluminum plate and were placed in a holder for the liquids (PEEK; diameter of 15 mm × 1.5 mm). A laser pulse was applied to the sample surface, and the resulting temperature response was recorded using an infrared detector. The obtained data were analyzed using the Proteus version 9 (Netzsch) to calculate the thermal diffusivity of the samples.

The thermal conductivity of the suspension was also predicted using the Maxwell-Garnett model to compare the theoretical predictions with measured values. In this study, it was assumed that the particles were uniformly dispersed and possessed a perfectly spherical shape. Therefore, no corrections were applied when the Maxwell-Garnett model was used. The effective thermal conductivities (*k_eff_*) of the suspensions were calculated using the following Maxwell-Garnett [17] model, as shown by Equation (5):(5)$$k_{eff}=k_{m}\left(\frac{k_{d}+2k_{m}-2\phi \left(k_{m}-k_{d}\right)}{k_{d}+2k_{m}+\phi \left(k_{m}-k_{d}\right)}\right)$$
where *k_d_*, *k_m_*, and ϕ represent the thermal conductivity (W·m^−^^1^·K^−^^1^) of the dispersed phase and matrix, and the volume fraction of the dispersed phase, respectively.

#### 2.5.3. Volume Expansion Coefficient

During the heat transfer of suspensions, the density change in the liquid continuous phase is significantly more significant than that of the solid dispersed phase, leading to buoyancy effects caused by the temperature changes that drive convection [18]. Therefore, the density of control change with temperature was measured to find the volumetric thermal expansion coefficient. The density was measured using a 10 mL pycnometer following the method described by Paul et al. [19]. The pycnometer was calibrated with distilled water at 25 °C and 50 °C. The target temperatures (25, 30, 40, 50, and 60 °C) were maintained in a thermostatic water bath. The density measurements at each temperature were repeated at least thrice. The temperature of the sample was monitored using an alcohol thermometer attached to a pycnometer, and the water bath temperature was measured using a digital thermometer installed on the device. The pycnometer was filled with the sample and placed in a thermostatic water bath without a lid until the desired temperature was attained. The lid was replaced after the desired temperature was reached. The pycnometer was quickly weighed using a balance with a sensitivity of 0.001 g. Density as a function of temperature (*ρ*(*T*)) can be fitted using the following exponential equation [20]; see Equation (6):(6)$$\rho \left(T\right)=\frac{\rho_{0}}{1+aT+bT^{2}+cT^{3}}$$
where *a*, *b*, and *c* are fitting constants optimized to match the experimental data. *ρ*_0_ is density at the reference temperature. *T* is temperature (C° or K).

The thermal expansion coefficient (*β*(*T*)) [19] is calculated as follows (7):(7)$$\beta \left(T\right)=-\frac{1}{\rho \left(T\right)}\frac{\partial \rho \left(T\right)}{\partial T}$$

By partially differentiating the temperature-density fitting equation from that of Sangwal [18] with respect to temperature and applying it to the thermal expansion coefficient equation, the thermal expansion coefficient as a function of temperature can be calculated by Equation (8):(8)$$\frac{\partial \rho \left(T\right)}{\partial T}=\frac{\partial }{\partial T}\left[\frac{\rho_{0}}{1+aT+bT^{2}+cT^{3}}\right]\;\frac{\partial \rho \left(T\right)}{\partial T}=-\frac{\rho_{0}\cdot \left(a+2bT+3cT^{2}\right)}{{\left(1+aT+bT^{2}+cT^{3}\right)}^{2}}$$

By substituting the ∂ρ(*T*)/∂T obtained above into the thermal expansion coefficient Equation (9):(9)$$\beta \left(T\right)=-\frac{1}{\rho \left(T\right)}\frac{\partial \rho \left(T\right)}{\partial T}$$

Thermal expansion coefficient derived from this process is as follows (10):(10)$$\frac{\partial \rho \left(T\right)}{\partial T}=\frac{\partial }{\partial T}\left[\frac{\rho_{0}}{1+aT+bT^{2}+cT^{3}}\right]\;\frac{\partial \rho \left(T\right)}{\partial T}=-\frac{\rho_{0}\cdot \left(a+2bT+3cT^{2}\right)}{{\left(1+aT+bT^{2}+cT^{3}\right)}^{2}}$$
where *a*, *b*, and *c* are fitting constants optimized to match the experimental data.

This equation reflects the nonlinear relationship between density and temperature, allowing for a straightforward calculation of the thermal expansion coefficient at a given temperature.

### 2.6. Natural Convection Heat Transfer Coefficient

#### 2.6.1. Natural Convective Heat Transfer Experimental System

The temperature change over time was measured under Rayleigh–Bénard convection conditions by heating only the bottom and insulating from external factors (Figure 1). The chamber was a polypropylene cylinder with a diameter of 4.8 cm and a height of 8 cm. The sample was then filled to 140 g. For the temperature measurements, T-type thermocouples were positioned at the center of the cylinder at heights of 1 cm, 4 cm, and 8 cm from the base. The cylinder was insulated with a 3 cm-thick layer of XPS. The cylinder was preheated to 30 °C in a water bath (WB-22, DAIHAN Scientific, Wonju, Republic of Korea) and then placed in a 40 °C water bath (SSB-45, DAIHAN Scientific, Wonju, Republic of Korea). Temperature was recorded using a data logger (MX100; Yokogawa, Tokyo, Japan).

#### 2.6.2. Natural Convective Dimensionless Number

Rayleigh (Ra) number is a dimensionless number that represents the strength of the natural convection flow caused by buoyancy in the fluid during natural convection heat transfer [21] and can be calculated using Equation (11):(11)$$Ra=\frac{\rho g\beta ∆TL^{3}}{\alpha \eta }$$
where *ρ*, *g*, *β*, ∆*T*, *L*, *α*, and *η* represent density, gravity acceleration (m·s^−2^), thermal expansion (K^−1^), temperature difference between the heating source and the sample (K), length (m), thermal diffusivity (m^2^·s^−1^), and viscosity (Pa·s), respectively.

Non-Newtonian fluids, like the samples in this study, make it challenging to define a specific shear rate for estimating viscosity [22]. The experimental environment involved a closed cylinder; the system exhibited axial symmetry. Therefore, heat flow is assumed to occur vertically. This experiment used the practical velocity scale from Ahmadi et al. [23], incorporating Herschel–Bulkley constants under Rayleigh–Bénard convection, to determine average fluid velocity (*U*) under natural convection. Equation (12) is as follows:(12)$$U=\sqrt[{n}]{\frac{\left(\rho g\beta ∆T-\tau_{0}\right)L^{n+1}}{K}}$$

Since the average shear rate $(\dot{\gamma_{e})}$ is the velocity gradient in the vertical direction of the fluid, it could be expressed in terms of the characteristic length and fluid velocity [24]. Equation (13) is as follows:(13)$$\dot{\gamma_{e}}=\frac{U}{L}$$

Kinematic viscosity was inferred using the Herschel–Bulkley model equation. The estimated average shear rate and kinematic viscosity were used to calculate *Ra_p_* values. The calculated *Ra_p_* value was substituted into the following Equation (14) to compute Nusselt numbers (Nu) [25]:(14)$$Nu=C\cdot {\left(Ra\right)}^{d}\;\;\;Ra<10^{4}:C=1.36,d=1/5\;10^{4}\leq Ra\leq 10^{9}:C=0.59,d=1/4\;Ra>10^{9}:C=0.13,d=1/3$$
where *C* and *d* are empirically determined constants for the vertical cylinder [25].

The convective heat transfer coefficient was calculated using the temperature variation data over time at different locations in the natural convection heat transfer experiment, and the corresponding Nue was derived. Rae was obtained by reversing the equation used to calculate Nu from Ra. The heat flow in the natural convection heat transfer experimental model was calculated using Equation (15):(15)$$q=m\times C_{p}\times \left(\frac{\Delta T_{s}}{\Delta t}\right)$$

The natural convection coefficient was calculated using the heat flow (*q*), surface area (*A*), and temperature difference between the heating source and the sample (∆*T*). Equation (16) is as follows:(16)$$h=\frac{q}{A\Delta T}$$

The Nusselt number can also be calculated using the natural convection coefficient (*h*), thermal conductivity (*k*), and characteristic length (*L*). Equation (17) is as follows:(17)$$Nu=\frac{h\cdot L}{k}$$

### 2.7. Statistical Analysis

All experiments were performed in triplicate to ensure reproducibility, and the results are expressed as mean values. Statistical analysis was conducted using SPSS Statistics software (version 24.0; IBM Corp., Armonk, NY, USA). To evaluate whether the differences among treatments were statistically significant, a one-way analysis of variance (ANOVA) was performed. When significant differences were detected, Duncan’s multiple range test was applied as a post hoc analysis to identify pairwise differences among treatment groups. A significant level of *p* < 0.05 was considered for all statistical tests.

## 3. Results and Discussion

### 3.1. Particle Size

The average particle sizes (D_[4,3]_) for four treatments were 120.25 µm, 31.35 µm, 10.48 µm, and 0.7548 µm, respectively (Table 1). The samples were labeled based on their rounded average particle sizes, with the sample homogenized at 28,620× *g* and rounded to the nearest thousand. Thus, the suspensions were named 120 (0.6–30), 31 (1–1780), 10 (3–7120), and 0.75 (3–28,620). To further assess particle distribution, particle sizes for the 120, 31, and 10 were measured using laser diffraction to compare volume and number fractions (Figure 2). Both 120 and 31 samples exhibited peaks in volume and number fractions smaller than the respective particle size. Notably, in the number fraction graph for the 120 (Figure 2C), the peaks for smaller particles and average-sized particles were 19.52% and 80.48%, respectively. Nevertheless, more than 99% of the total integrated areas in volume fraction graphs corresponded to the peaks representing the average particle sizes in both samples. This observation is significant because volume fraction, more than number fraction, influences thermal and rheological properties [26,27].

The volume fraction was concentrated around the average particle size in all the samples. Therefore, the average particle size for each suspension was considered suitable for characterizing the rheological and thermal properties discussed in this study.

### 3.2. Polarization Microscopy Images

Polarization microscopy images of the suspensions are presented in Table 2. All samples exhibited particles with a spherical shape, with particle size decreasing as the homogenization speed increased. The spherical morphology of stearic acid particles is likely due to the emulsion state prior to crystallization [28]. According to Gram et al. [29], the shape of dispersed particles plays a key role in determining the rheological properties of a suspension. Consequently, the spherical particles observed in this study were considered to effectively regulate particle morphology variables, thereby influencing the rheological behavior of the suspensions.

### 3.3. Rheological Properties

The change in apparent viscosity with shear rate for the suspensions and control is shown in Figure 3. All samples, including the control, exhibited shear-thinning behavior. At a low shear rate (below 17/s), the control exhibited the lowest apparent viscosity, while suspension with smaller particles demonstrated higher viscosities. As the shear-thinning effect intensified with smaller particles, the order of apparent viscosity above 87/s changed to 0.75, 10, 31, 120, and control.

Table 3 summarizes the rheological parameters for each sample, determined by fitting the flow curve data using the Herschel–Bulkley model. The consistency index reflects the viscosity of a fluid, with a higher value indicating greater viscosity [29]. The flow index, on the other hand, indicates the fluid’s response to shear deformation. Newtonian fluids exhibit an n value of 1, shear-thickening fluids exhibit *n* > 1, and shear-thinning fluids exhibit *n* < 1. A smaller n value corresponds to a more rapid decrease in viscosity as the shear rate increases [19].

All samples in this study exhibited flow indices below 1, confirming their shear-thinning behavior. The control sample had the lowest yield stress, consistency index, and the highest flow index. These values were consistent with previous reports for xanthan gum solutions of the same concentration [20], where the yield stress ranged between 0.9–1.5 Pa, the consistency index between 0.45–0.8 Pa∙s^n^, and the flow index between 0.43–0.64.

The yield stress values for 120, 31, and 10 were higher than that of the control but not significantly different (*p* > 0.05). However, 0.75 exhibited a significantly higher yield stress than all other samples (*p* < 0.05). Regarding the consistency index, only sample 120 did not show a significant difference compared to the control (*p* > 0.05), although its value was still higher. Samples 31, 10, and 0.75 had significantly higher consistency indices than the control, with smaller particle sizes correlating with higher consistency indices (*p* < 0.05).

Similarly, the flow indices of 31, 10, and 0.75 were significantly different from those of the control, with smaller particle sizes corresponding to lower flow index (*p* < 0.05). As the particle size decreases, the surface area increases, leading to more uniform particle dispersion and greater resistance to flow. This increase in surface area leads to enhanced particle–particle and particle–medium interactions, which results in higher viscosity, yield stress, and consistency index [21]. However, as the shear rate increases, particle–particle interactions such as hydrodynamic interactions, repulsive, attractive forces, and contact forces tend to decrease. Smaller particles experience a greater reduction in these interactions due to their larger surface areas, which explains the observed reduction in flow resistance at higher shear rates [30,31,32].

In Figure 4, both G′ and G″ decreased with increasing temperature for all samples, reflecting the thermal softening behavior typical of viscoelastic fluids. The control sample, which lacked stearic acid particles, exhibited the lowest moduli across the temperature range. In contrast, stearic acid-containing suspensions showed higher G′ and G″ values, with the moduli increasing with particle size. This can be attributed to the filler effect of stearic acid particles, which reinforce the particle–medium network and enhance structural integrity. The temperature-induced decline in both moduli is likely due to the reduced viscosity and weakened molecular entanglements of the continuous phase (xanthan gum solution), resulting from increased molecular mobility. Larger particle sizes resulted in higher G′ and G″ values, likely due to increased particle–particle interactions and physical entanglements within the suspension network. These structural interactions facilitate greater energy storage and viscous dissipation during deformation. The relatively moderate changes in G′ and G″ with temperature are likely due to the high thermal stability of xanthan gum, which preserves its network structure even at elevated temperatures [33].

### 3.4. Thermal Properties

#### 3.4.1. Specific Heat Capacity

The specific heat capacities of the control and stearic acid suspensions are shown in Figure 5A. The control exhibited an increasing trend in specific heat with temperature and no significant differences in the specific heat values among the suspensions (*p* > 0.05). However, the control showed significantly higher values than the suspensions (*p* < 0.05). To verify the accuracy of the experimental values, we predicted the specific heat values at various temperatures and compared them with measured data. Muthamizhi et al. [34] reported a reliable equation (R^2^ = 0.9774) for predicting the specific heat of xanthan gum based on concentration and temperature:
*C_p_* = 98.78575 − 0.6495*T* − 15.9955*X* + 0.06*TX* + 0.001109*T*^2^ − 0.5*X*^2^

The predicted specific heat values of the continuous phase (control) were calculated using this equation and plotted against temperature in Figure 5A. The specific heat values of the suspensions were predicted using the rule of mixtures [35]:
*C_p eff_* = *w_d_*·*C_pd_* + (1 − *w_d_*)·*C_p con_*

The specific heat values of stearic acid were obtained from previously reported data [35]. Figure 5A shows the predicted and measured values for the control and suspensions. As the temperature increased, the predicted and measured values exhibited a growing deviation from the predicted values. Below 35 °C, there was no significant difference from the predicted and measured specific heat values for suspensions (*p* > 0.05). However, above 40 °C, the specific heat values of the suspensions were significantly higher than the predicted values (*p* < 0.05).

This deviation at higher temperatures suggests that thermal resistance at the particle surface may have increased, leading the system to absorb more thermal energy, as suggested by Wang et al. [36] and Carrillo-Berdugo et al. [37]. Conversely, other studies suggest that interfacial thermal resistance may lower heat transfer, reducing the specific heat [38,39]. In this study, the lack of significant differences among the suspensions suggests that the surface area was insufficient to meaningfully alter the specific heat of the system [40].

#### 3.4.2. Thermal Conductivity

The thermal conductivities of the control and suspended samples are presented in Table 4. The thermal conductivities of the control sample containing no stearic acid particles were 0.598 W/m∙K and 0.684 W/m∙K at 25 °C and 40 °C, respectively, exhibiting the highest values at all measured temperatures.

Additionally, for all suspension samples, the thermal conductivity increased as the particle size decreased at all temperatures. However, except for 40 °C, there were no significant differences in the thermal conductivity between the control and suspension samples at the measured temperatures (*p* > 0.05). At 40 °C, the thermal conductivity of sample 120 (0.663 W/m∙K) was significantly different from that of the control sample (*p* < 0.05), whereas the thermal conductivities of 31 (0.671 W/m∙K), 10 (0.675 W/m∙K), and 0.75 (0.679 W/m∙K) exhibited no significant difference compared to that of the control (*p* > 0.05). This trend suggests that the thermal conductivity of samples with smaller particles increases more rapidly as the temperature increases.

The predicted values for the suspensions, calculated using the Maxwell-Garnett model based on the thermal conductivity of the control sample, and the significance of the differences between the predicted values and the suspension samples are also listed in Table 4. The thermal conductivities predicted for the 3 wt.% stearic acid suspension using the Maxwell-Garnett model were 0.590 W/m∙K, 0.619 W/m∙K, 0.646 W/m∙K, and 0.672 W/m∙K at 25 °C, 30 °C, 35 °C, and 40 °C, respectively. No significant differences were observed between the predicted values of the suspension samples and their thermal conductivities at any of the measured temperatures (*p* > 0.05).

The effect of particle size reduction on the thermal conductivity of suspensions has been extensively reported in studies investigating metallic nanoparticles [41,42]. A well-known mechanism for enhancing the thermal conductivity of nanoparticles is the efficient heat transfer at the interface, provided by the increased surface area resulting from their smaller particle size [41]. Another significant factor is the formation of new heat transfer pathways owing to particle collisions through Brownian motion within the continuous phase [42]. Additionally, if the particles exhibit a higher thermal conductivity than the continuous phase, they facilitate faster heat transfer. However, for micro-sized particles, such as those used in this study, the Brownian motion of the particles is limited or nonexistent, thus restricting the mechanisms that enhance thermal conductivity. According to Cherkasova [43], both nanosized and micro-sized particles exhibit an increase in thermal conductivity as the particle size decreases; however, the effects differ. Nanosized particles exhibited an enhancement in thermal conductivity that exceeded the predictions of the Maxwell-Garnett model. In contrast, the thermal conductivity of suspensions with micro-sized particles closely matched the values predicted by the model. Although smaller particles exhibit higher thermal conductivity, the lack of significant deviation from the Maxwell-Garnett model suggests that the effect of particle size on the thermal conductivity of micro-sized particles may be relatively minor [44]. This insensitivity indicates that factors such as particle–particle interactions, interfacial thermal resistance, and Brownian motion, which influence thermal conductivity, may not have had a significant impact when micro-sized particles are dispersed in the suspension [43]. Moreover, stearic acid, used as the dispersed phase in this study, exhibited a lower thermal conductivity (0.35 W/m·K) than the xanthan gum solution used as the continuous phase, which may have hindered the improvement in thermal conductivity due to the increased surface area.

Furthermore, stearic acid is hydrophobic, which probably causes interfacial thermal resistance in the polar xanthan gum solution, reducing thermal conductivity [44]. Despite these limitations, a trend of higher thermal conductivity with smaller particles was consistently observed at all the measured temperatures, albeit with slight differences. At 40 °C, a significant difference in thermal conductivity was observed between sample 120, which possessed the largest particles, and the other samples (*p* < 0.05). Particle surfaces in suspensions form new, thin thermal boundary layers that enhance thermal conductivity through mechanisms such as shear flow, particle–fluid interactions, and the formation of liquid layers. Lenin et al. [45] also reported that as the particle size of stearic acid-coated particles decreased, the thermal resistance due to the low thermal conductivity of stearic acid increased, ultimately resulting in a lower rate of increase in thermal conductivity compared to suspensions with uncoated particles. Nevertheless, the smaller particles exhibited higher thermal conductivities among the suspensions with stearic acid-coated particles. Additionally, the samples in this study employed a relatively high-viscosity continuous phase, consisting of a 0.5% xanthan gum solution. High viscosity can limit the impact of particle size reduction on thermal conductivity by increasing the interfacial thermal resistance between the dispersed and continuous phases, restricting particle interactions, and limiting particle movement [46]. In conclusion, the samples in this study, which modeled high-viscosity suspensions with a thickening agent and a continuous phase of low thermal conductivity, did not exhibit significant deviations from the Maxwell-Garnett model predictions, regardless of particle size. Moreover, except at 40 °C, there were no significant differences in the thermal conductivity between the control and suspension samples. However, at all measured temperatures, there was a consistent trend of higher thermal conductivity with smaller particles, suggesting that while particle size reduction enhanced the thermal conductivity, the effects were limited by the relatively high viscosity of the samples and the properties of the particles that had lower thermal conductivity than that of the continuous phase.

#### 3.4.3. Volumetric Thermal Expansion Coefficient

The temperature-dependent densities of water and the control, along with the nonlinear fitting curve for the control, are shown in Figure 5B. The density of the control from 1.000 g/cm^3^ at 25 °C to 0.985 g/cm^3^ at 60 °C, closely following the trend for water. This pattern is consistent with the results of Dhiaa [47], who measured the density and viscosity of xanthan gum across various concentrations at different temperatures. The nonlinear fitting of the measured density values as a function of temperature yielded a high reliability (R^2^ = 0.9998), confirming the accuracy of the data [20]. The calculated thermal expansion coefficients were 1.000/°C at 25 °C and 0.990/°C at 50 °C, with the values from 25 °C to 60 °C are shown in Figure 5C.

### 3.5. Dimensionless Numbers for Natural Convection

#### 3.5.1. Rayleigh Number

The changes in the dimensionless numbers for the natural convection heat transfer at different temperature measurement points (1, 4, and 8 cm above the center of the bottom surface) and the dimensionless numbers predicted by theoretical calculations are presented in Figure 6 and Figure 7. When Ra exceeds Ra_c_, the fluid converts into a stable laminar flow. As Ra increases, the viscosity of the fluid decreases, allowing the heated fluid to ascend more rapidly and thereby intensify natural convection. Additionally, an increase in Ra signifies a reduction in the thickness of the thermal boundary layer due to the steepened temperature gradient near the heated region, where heat transfer is concentrated. Finally, higher Ra values result in more substantial and more vigorous circulation of convection cells, leading to enhanced upward heat transfer [48]. All samples in this study exceeded Ra values of 10^5^ at the beginning of heating (Figure 6), which is considered very high. Thus, it can be inferred that convection began in the early heating stages, allowing the yield stress to be disregarded [48]. The Ra_c_ for Rayleigh–Bénard convection in shear-thinning fluids is approximately 1700 n^2^, with values calculated by substituting the n values for each sample. The calculated Ra_c_ is indicated as a baseline on the Ra graph in Figure 6.

In the case of Ra_p_ (Figure 6A), the values exceeded those of Ra_c_ across all the temperature ranges. Indicating that convection occurred at all temperature differences. For Ra_p_, however, there were temperature differences at all measurement positions where the values were below all Δ*T*. For the suspensions at Δ*T* = 10 °C, the Ra_p_ values were in descending order: 0.75, 10, 120, and 31. As Δ*T* decreased, the ranking reversed, with 120 showing the highest Rap value at Δ*T* = 1 °C.

At 1 cm, closest to the heated surface, Ra_e_ (Figure 6B) showed higher values than Ra_p_. At Δ*T* of 9 °C, the Ra_e_ of the control was the highest, with smaller particle suspensions showing higher than the control. For suspensions at the same Δ*T*, the Ra_e_ values were higher as particle size decreased. At 1 cm, starting from a Δ*T* of 5 °C, 0.75 exhibited a Ra_e_ value, which exceeded that of the control. As Δ*T* increased, the difference between the two values increased. At Δ*T* = 1 °C, the Ra_e_ of 0.75 was more than double that of control. In summary, smaller particles generally exhibited higher Ra_e_ values, and the Δ*T* at which convection ceased (Ra_e_ < Ra_c_) decreased with smaller particle sizes.

At 4 cm (Figure 6C), the Ra_e_ at a Δ*T* of 9 °C followed order: the control, 0.75, 10, 120, and 31. The Ra_e_ values at 4 cm were considerably smaller than those at 1cm. Among the suspensions, 0.75 exhibited the highest Ra_e_, except at Δ*T* = 1 °C. However, unlike at 1 cm, 31 showed smaller Ra_e_ values than 120, despite having smaller particles, indicating that particle size did not always correlate with higher Ra_e_ in this position.

At 8 cm, the furthest point from the heating surface (Figure 6D), the differences in Ra_e_ between the samples were smaller. At Δ*T* = 9 °C, all samples had Ra_e_ values roughly ten times smaller than those observed at 4 cm. All samples exhibited Ra_e_ values below Ra_c_ when Δ*T* was approximately 5 °C. The differences in Ra_e_ between the samples were minimal, indicating that particle size had less impact on convection at this distance.

The rapid decrease in initial Ra_e_ values may indicate the inclusion of other variables, such as turbulence, during early heating when Δ*T* is large. Weak turbulence typically begins at an Ra of approximately 10^5^, with fully developed turbulence occurring above 10^7^ in Rayleigh–Bénard convection [21]. There were differences in the Ra values and trends between the Ra_p_ and Ra_e_ for each sample. The average velocity equation that predicted Ra_p_ only considers buoyancy and fluid behavior characteristics due to temperature differences in Rayleigh–Bénard convection [24]. Consequently, the accuracy of the predictions was inevitably limited as the variations in convective intensity due to temperature gradient changes at each measurement location were not considered. At 1 cm and 4 cm, when the Δ*T* values were approximately 4–5 °C and 8 °C, respectively, the Ra value reached 10^5^, indicating that convection of similar intensity occurred within this Δ*T* range. In contrast, at 8 cm, the point furthest from the heating surface, Ra_e_ temporarily exceeded 10^5^ at the initial heating stage when Δ*T* was 9 °C, whereas the value for 10 was 8.48 × 10^4^, below 10^5^. At 8 cm, the differences in Ra values between samples were minimal compared to other conditions. Samples with lower n values, corresponding to smaller particles, experienced greater shear-thinning as fluid velocity increased due to buoyancy, which accelerated convective heat transfer [49]. This explains why smaller particles, such as 10 and 0.75, exhibited higher Ra_e_ values at 1 and 4 cm, with a steeper temperature gradient. Conversely, at 8 cm, where the temperature gradient is lower, samples with smaller particles exhibited lower Ra values compared to those with relatively larger particles due to the relatively weaker convective intensity. This indicates slower fluid flow, resulting in less shear-thinning behavior. In this case, the effects of K and τ_0_ become more pronounced as the smaller K and τ_0_ values for larger particle samples reduce the resistance to flow relative to other samples. Consequently, these samples exhibit higher Ra values, indicating stronger convective intensity. Shear-thinning fluids exhibit a decrease in viscosity with increasing shear rate, which promotes fluid mobility and enhances the movement of convection cells. These effects are particularly significant under strong temperature gradients, which induce spatial variations in viscosity. In heated regions, the viscosity decreases substantially, facilitating the formation of upward plumes. In contrast, the viscosity remains relatively high in cooler regions, thereby limiting downward flow. This asymmetric viscosity distribution contributes to the directionality and intensity of Rayleigh–Bénard convection [50]. In this study, suspensions containing smaller particles demonstrated more pronounced shear-thinning behavior, as indicated by lower flow index values. Correspondingly, higher Rayleigh numbers were observed under large temperature differentials (Δ*T*). These findings suggest that particle size affects convective heat transfer not only by altering flow resistance but also by modulating the temperature-dependent viscosity profile in shear-thinning systems, ultimately enhancing the efficiency of natural convection [51].

#### 3.5.2. Nusselt Number

Nu has an inverse relationship with the thickness of the thermal boundary layer. Therefore, a higher Nu indicates a thinner thermal boundary layer. In the case of Nu_p_, it allows for estimating the average thickness of the thermal boundary layer across the entire sample, providing insight into the overall heat transfer activity. On the other hand, Nu_e_ at each measurement location reflects the local thermal boundary layer conditions, specifically indicating the magnitude of the actual temperature gradient at that point [52]. Nu values close to 1 indicate that conduction dominates heat transfer, while Nu values between 2 and 10 suggest a mixed conduction–convection region [53]. When Nu exceeds 10, it indicates convection dominates [23]. For Nu_p_, all samples exhibited values exceeding 3, with values close to 10 up to Δ*T* = 5 °C, suggesting that convection dominated heat transfer across all Δ*T* values. The minimal differences in Nu_p_ between samples suggest that similar convective heat transfer efficiency and thermal boundary layer thickness were achieved during heating. Nu_e_ exhibited some differences in trends compared with Nu_p_, depending on the measurement point. Even at 1 cm, the point closest to the heating surface (Figure 7A,B), the highest Nu_e_ value at Δ*T* = 1 °C, observed for 0.75, was 1.95, indicating a weakening of the convective heat transfer. The lowest Nu_e_ value observed for 31 was 1.25, which is a very low value, suggesting that most of the heat transfer shifted to conduction [49].

At 1 cm with an Δ*T* of 9 °C (Figure 7B), all samples exhibited Nu_e_ values exceeding 100, indicating that a sufficiently strong temperature gradient was established near the heated surface to generate intense convection [52]. Smaller particles like 10 and 0.75 were more affected by convection, showing higher Nu_e_ values. As Δ*T* decreased, the Nu_e_ values for 120 and 31 fell below 10 at Δ*T* = 4 °C, suggesting weaker convection. All samples showed Nu_e_ values between 2 and 10 from Δ*T* = 5 to 2 °C, indicating convection-dominated heat transfer in this range.

At 4 cm (Figure 7C), unlike the initial heating stage at 1 cm where Nu_e_ values exceeded 100, the Nu_e_ values at Δ*T* = 9 °C ranged between 20 and 30. This can be attributed to the location being 3 cm away from the heating region, where the temperature gradient is smaller [53]. The Nu_e_ values decreased as Δ*T* decreased. At Δ*T* = 9 °C, the control sample and the 0.75 sample exhibited the highest Nu_e_ values, but differences between samples were minimized when Δ*T* < 5 °C. The Ra_e_ value of 31 was consistently lower than that of 120, indicating that at this position, the temperature gradient, thermal boundary layer thickness, and convective effects do not exhibit a proportional relationship with particle size.

At 8 cm (Figure 7D), differences in Nu_e_ values between samples were minimized across all Δ*T* values, with no sample showing a difference greater than 1. All samples exhibited Nu_e_ values below 2 at Δ*T* = 2 °C, indicating reduced convective efficiency at this distance. Furthermore, this suggests that the differences in thermal boundary layer thickness and temperature gradient among the samples were less pronounced compared to 1 cm and 4 cm [52]. Notably, smaller particles, such as 0.75 and 10, showed lower Nu_e_ values compared to 120 and 31 when Δ*T* < 5 °C, indicating that smaller particles had lower heat transfer efficiency at this location [53].

These results indicate that the effect of particle size on convective heat transfer efficiency depends on the temperature gradient and the distance from the heating surface. Under high temperature differentials near the heated region, smaller particles tended to exhibit higher Nusselt numbers due to more pronounced shear-thinning behavior. In contrast, at lower temperature differences or positions farther from the heat source, larger particles showed relatively higher Nu, suggesting that thermal boundary layer development and local viscosity variations play a critical role in determining heat transfer performance.

In theoretical predictions, Ra_p_ and Nu_p_ failed to accurately capture the trends of Ra_e_ and Nu_e_ because they do not account for the significant temperature gradient conditions that occur closer to the heated region during the initial heating stage. However, the trends in the dimensionless numbers with respect to Δ*T* for the samples aligned with some of the experimental trends. For example, at 1 cm, the Rae for the 31 samples was larger than that for the 120 samples at most Δ*T* values. However, at 4 cm and 8 cm, both Ra_p_ and Nu_p_ for the 31 samples were lower than those for the 120 samples, with minimal differences between the two samples. This is attributed to the rheological properties associated with particle size. The 120 samples exhibited a lower consistency index but a higher flow index, meaning that the decrease in viscosity with increasing shear rate was slower (*p* < 0.05). Therefore, depending on the heating conditions, the relative flow velocities of these two samples could be reversed [22]. Additionally, the experimental and theoretical predictions showed that as Δ*T* decreased, smaller particles experienced less shear-thinning, leading to a rise in viscosity and a slower rate of natural convection heat transfer. When Δ*T* reached 4 °C, larger particle samples exhibited higher Ra and Nu values, reversing the trends associated with particle size. This phenomenon suggests that as Δ*T* decreases, the buoyancy effects caused by the temperature difference diminish, leading to smaller particles returning to higher viscosity and ultimately slowing down natural convection [20]. Furthermore, in suspensions with smaller particles, although K and τ_0_ are larger, the smaller n value results in greater flow resistance as convective intensity decreases. This can accelerate the thickening of the thermal boundary layer compared to suspensions with larger particles [50].

## 4. Conclusions

This study examined the effect of particle size on rheological and thermal properties and on convection heat transfer in a high-viscosity food suspension model. The experimental results demonstrated that smaller particle sizes increased the suspension’s thermal conductivity and shear-thinning behavior. Regarding natural convective heat transfer, samples with smaller particles exhibited higher Ra and Nu values, particularly at higher shear rates. However, larger particles displayed higher Ra and Nu values as the temperature difference between the sample and the heated surface decreased along with shear. These results are believed to be due to rheological changes under the heating environment, along with corresponding variations in the temperature gradient and thermal boundary layer. These results suggest that fine particles can potentially improve heat transfer efficiency in high-viscosity suspensions, emphasizing the importance of optimizing particle size and heating parameters. Understanding how fine particles behave in viscous media is essential for improving heat treatment in real food processes. The current model was designed under controlled conditions, which may not fully reflect the complexity of real systems. Future research should consider more realistic particle types, flow conditions, and system designs to improve practical relevance.

## Figures and Tables

**Figure 1 foods-14-02625-f001:**
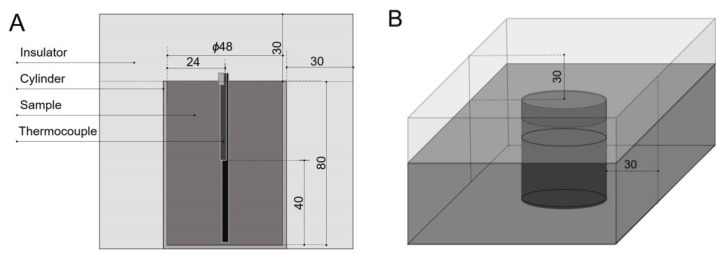
Cross-sectional (**A**) and perspective (**B**) views of the natural convection heat transfer test module.

**Figure 2 foods-14-02625-f002:**
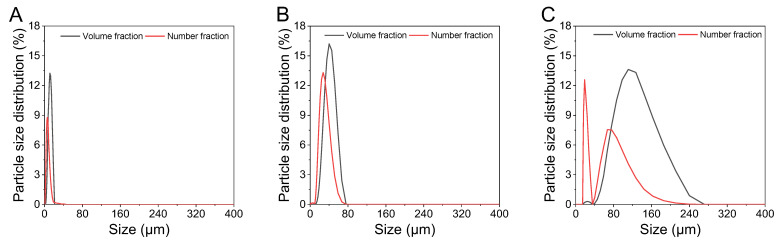
Particle size distribution curves of stearic acid suspension ((**A**): 120, (**B**): 31, (**C**): 10) measured using a laser diffraction spectrometer analyzer. The labels 120, 31, and 10 refer to the average particle size rounded to one decimal place.

**Figure 3 foods-14-02625-f003:**
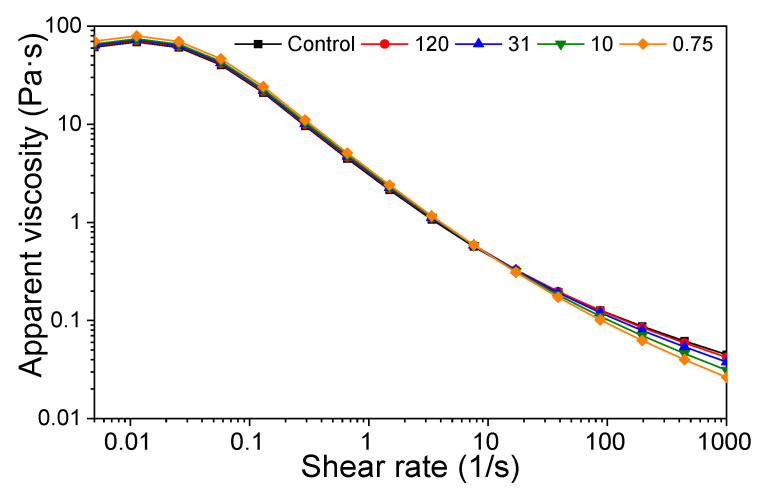
Apparent viscosity of control and stearic acid suspension samples. The labels 120, 31, and 10 refer to the average particle size rounded to one decimal place, while 0.75 was rounded to three decimal places.

**Figure 4 foods-14-02625-f004:**
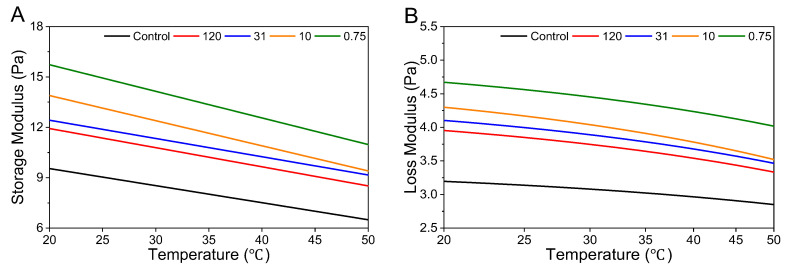
Storage (**A**), and loss modulus (**B**) of control and stearic acid suspension samples. The labels 120, 31, and 10 refer to the average particle size rounded to one decimal place, while 0.75 was rounded to three decimal places.

**Figure 5 foods-14-02625-f005:**
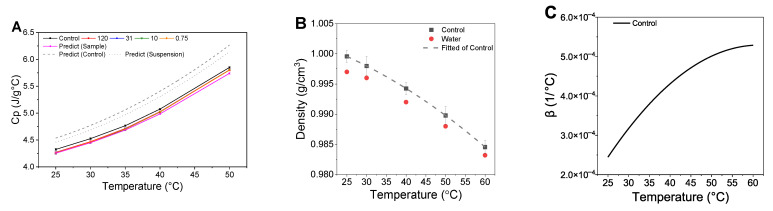
Specific heat capacity (**A**), density of the control (**B**), and thermal expansion coefficient of the control (**C**). Cp represents the specific heat capacity, and β represents the thermal expansion coefficient of the control. The labels 120, 31, and 10 correspond to the average particle sizes rounded to one decimal place, while 0.75 was rounded to three decimal places. Predict (control) indicates the predicted specific heat of the control using the equation by Muthamizhi et al. [34]. Predict (sample) refers to the predicted specific heat of the suspensions calculated using the rule of mixtures and measured specific heat of the control. Predict (suspension) represents the specific heat of the suspensions, calculated with the rule of mixtures using the predicted values of Predict (control).

**Figure 6 foods-14-02625-f006:**
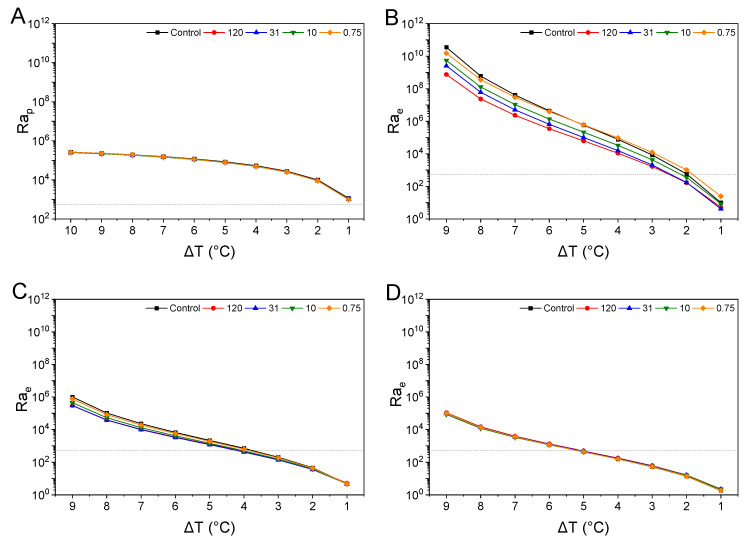
The predicted Rayleigh number (**A**) was based on the theoretical mean velocity derived from the temperature difference (Δ*T*), and Rayleigh numbers calculated based on Δ*T* between the heating temperature and the sample temperature at 1 cm (**B**), 4 cm (**C**), and 8 cm (**D**) vertically from the center of the bottom surface. Ra_p_ represents the predicted Rayleigh number, and Ra_e_ represents the calculated Rayleigh number. The labels 120, 31, and 10 correspond to the average particle sizes rounded to one decimal place, while 0.75 was rounded to three decimal places.

**Figure 7 foods-14-02625-f007:**
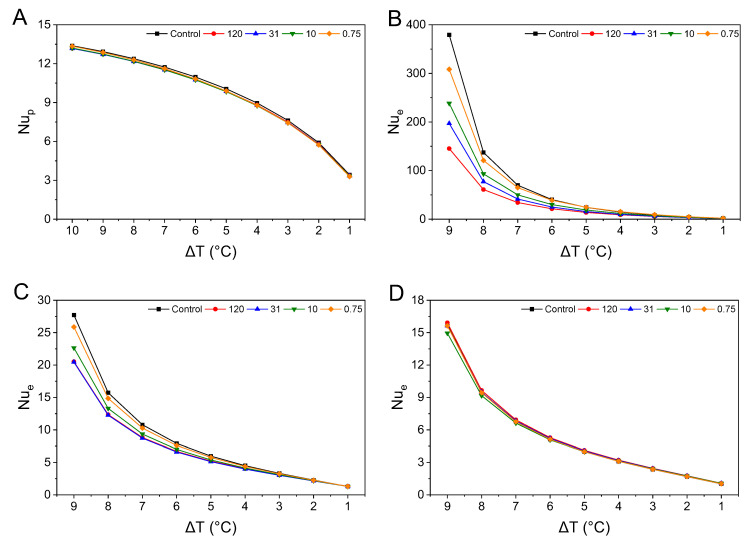
The predicted Nusselt number (**A**) was based on the theoretical mean velocity predicted by the temperature difference (Δ*T*), and Nusselt number calculated based on Δ*T* between the heating temperature and the sample temperature at 1 cm (**B**), 4 cm (**C**), and 8 cm (**D**) vertically from the center of the bottom surface. Nu_p_ represents the predicted Nusselt number, while Nu_e_ represents the calculated Nusselt number. The labels 120, 31, and 10 correspond to the average particle sizes rounded to one decimal place, while 0.75 was rounded to three decimal places.

**Table 1 foods-14-02625-t001:** Average particle sizes of stearic acid suspensions.

Treatments ^1^	Condition ^2^	D_[4,3]_ (μm)	Span
120	0.6–30	120.25 ± 0.45	1.14 ± 0.35
31	1–1780	31.35 ± 1.32	1.15 ± 0.08
10	3–7120	10.48 ± 0.30	1.63 ± 0.02
		**Z-Average (nm)**	**PDI**
0.75	3–28,620	754.80 ± 55.12	0.45 ± 0.12

^1^ Each treatment received a label through rounding the average particle size to one decimal place (0.75 was rounded to three decimal places). ^2^ Conditions are labeled according to the ratio of Tween^®^ 80 (wt%) and homogenization speed (*×g*) used to create the stearic acid emulsion.

**Table 2 foods-14-02625-t002:** Polarization microscopy images of suspensions at various homogenization speeds.

Treatments ^1^	120	31	10	0.75
Polarization microscopy images	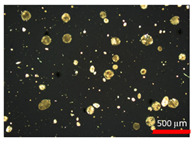	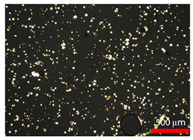	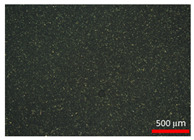	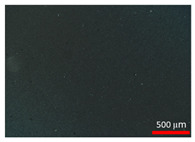

^1^ Each treatment received a label through rounding the average particle size to one decimal place (0.75 was rounded to three decimal places).

**Table 3 foods-14-02625-t003:** Herschel–Bulkley parameters and flow behavior for each suspension.

Treatments ^1^	*τ*_0_ (Pa)	*K* (Pa∙s^n^)	*n*	R^2^
Control	1.93 ± 0.25 ^b^	0.56 ± 0.08 ^d^	0.63 ± 0.02 ^a^	0.997
120	1.96 ± 0.26 ^b^	0.62 ± 0.09 ^d^	0.60 ± 0.02 ^a^	0.996
31	1.97 ± 0.27 ^b^	0.66 ± 0.11 ^c^	0.58 ± 0.02 ^b^	0.995
10	1.98 ± 0.28 ^b^	0.73 ± 0.12 ^b^	0.53 ± 0.03 ^bc^	0.993
0.75	2.07 ± 0.29 ^a^	0.75 ± 0.14 ^a^	0.50 ± 0.03 ^d^	0.992

^1^ Each treatment received a label through rounding the average particle size to one decimal place (0.75 was rounded to three decimal places). ^a–d^ Means labeled with different letters in the same column indicate a significant difference at *p* < 0.05 using Duncan’s multiple range test. *τ*_0_ is apparent yield stress, *K* is consistency index, and *n* is flow index.

**Table 4 foods-14-02625-t004:** Thermal conductivity of control, suspension samples and predicted values of suspension sample from the Maxwell-Garnett model.

Treatments ^1^	Thermal Conductivity (w/m·K)			
	Temperature (°C)			
	25	30	35	40
**Control**	0.598 ± 0.015 ^b^	0.628 ± 0.011 ^b^	0.657 ± 0.018 ^b^	0.684 ± 0.008 ^a^
**120**	0.586 ± 0.013 ^a^	0.611 ± 0.012 ^a^	0.639 ± 0.008 ^a^	0.663 ± 0.009 ^a^
**31**	0.590 ± 0.015 ^b^	0.615 ± 0.012 ^b^	0.645 ± 0.014 ^b^	0.671 ± 0.011 ^ab^
**10**	0.592 ± 0.009 ^b^	0.619 ± 0.014 ^b^	0.648 ± 0.009 ^b^	0.675 ± 0.010 ^ab^
**0.75**	0.595 ± 0.010 ^b^	0.624 ± 0.012 ^b^	0.652 ± 0.012 ^b^	0.679 ± 0.008 ^ab^
**MG ^2^**	0.590 ± 0.014 ^b^	0.619 ± 0.016 ^b^	0.646 ± 0.017 ^b^	0.672 ± 0.008 ^ab^

^1^ Each treatment received a label through rounding the average particle size to one decimal place (0.75 was rounded to three decimal places). Control consists solely of a continuous phase. ^2^ MG represents the thermal conductivity values predicted by the Maxwell-Garnett model. ^a,b^ Means labeled with different letters in the same row indicate a significant difference at *p* < 0.05 using Duncan’s multiple range test.

## Data Availability

The original contributions presented in this study are included in the article. Further inquiries can be directed to the corresponding authors.

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
