# Peer review of "Mathematical Modeling and Microparticle Size Control for Enhancing Heat Transfer Efficiency in High-Viscosity Food Suspensions"

_foods, 2025, doi:10.3390/foods14152625_

Round 1

Reviewer 1 Report

Comments and Suggestions for Authors

Major Comments:

1.The rationale behind selecting stearic acid as the dispersed phase should be clarified in the manuscript.

2. For a clear understanding of the limitations of the results of this research. The decision to use 3 wt.% stearic acid across all treatments might benefit from justification regarding its representativeness for real-world formulations.
3. In view of the fact that food processing often involves heating procedures, it is recommended to add the analysis of the impact of temperature changes on fluid G' and G" in rheological analysis
4. The interplay between particle size, viscosity, and flow-induced heat transfer is complex. Could the authors expand on the mechanistic hypothesis, particularly regarding how shear-thinning behavior modulates Rayleigh-Bénard convection?

5. Does Zeta Potential affect rheology?
Minor Comments:
1. The line number is missing in the manuscript.
2. The parameters expression is suggested using × g instead of rpm.
3. The purity information of the reagent should be mentioned.
5. Make sure that the meaning of each symbol in the formula is clearly stated as much as possible

Reviewer 2 Report

Comments and Suggestions for Authors

Comments and suggestions for the authors are provided in the attached file.

Reviewer 3 Report

Comments and Suggestions for Authors

The manuscript titled “Mathematical Modeling and Microparticle Size Control for Enhancing Heat Transfer Efficiency in High-Viscosity Food Suspensions” is well-organized, methodologically strong, and addresses an important research gap. The clarity of the experimental design and its integration with theoretical models are commendable. Minor improvements could include:

  • The abstract lacks quantitative results.

  • The keyword list is adequate but could be expanded to include terms like "rheology" and "thermal conductivity" for broader indexing.

  • What is the reason for using xanthan gum and stearic acid to obtain experimental results?

  • What is Twen® 80? Please provide an explanation.

  • To which natural food model could this experimental model be applied? Possibly to chocolate, considering that it behaves like a non-Newtonian fluid?

  • What limits the experimental model? Aside from temperature or, more specifically, the temperature regime?

  • Please update the list of references, especially those related to stearic acid. The authors have only cited one from 1950.

Round 2

Reviewer 1 Report

Comments and Suggestions for Authors

The author has made the required revisions to the paper and answered all my doubts. The quality of the article has greatly improved, and it is recommended to accept it